# Investigation of Bunch Compressor and Compressed Electron Beam Characteristics by Coherent Transition Radiation

**Sikharin Suphakul [1],\*, Heishun Zen [2], Toshiteru Kii [2] and Hideaki Ohgaki [2]**

[1]  Plasma and Beam Physics Research Facility, Chiang Mai University, Chiang Mai 50200, Thailand
[2]  Institute of Advanced Energy, Kyoto University, Kyoto 611-0011, Japan; zen@iae.kyoto-u.ac.jp (H.Z.);
    kii@iae.kyoto-u.ac.jp (T.K.); ohgaki.hideaki.2w@kyoto-u.ac.jp (H.O.)
\*  Correspondence: sikharin.sup@gmail.com; Tel.: +66-(0)-91-850-8896

**Abstract:** A magnetic chicane bunch compressor for a new compact accelerator-based terahertz (THz) radiation source at the Institute of Advanced Energy, Kyoto University, was completely installed in March 2016. The chicane is employed to compress an electron bunch with an energy of 4.6 MeV generated by a 1.6-cell photocathode radio frequency (RF)-gun. The compressed bunch is injected into a short planar undulator for THz generation by coherent undulator radiation (CUR). The characteristics of the bunch compressor and the compressed bunch were investigated by observing the coherent transition radiation (CTR). The CTR spectra, which were analyzed by using a Michelson interferometer, and the compressed bunch length were also estimated. The results were that the chicane could compress the electron bunch at a laser injection phase less than 45 degrees, and the maximum CTR intensity was observed at a laser injection phase around 24 degrees. The optimum value of the first momentum compaction factor was around $-45$ mm, which provided an estimated rms bunch length less than 1 ps.

**Keywords:** coherent transition radiation; magnetic bunch compressor; electron bunch length

## 1. Introduction

Small-scale electron accelerator systems have been proven to have the potential to produce a high power and tunable radiation in the range of far-infrared and terahertz (THz) regimes. These features are significantly beneficial to THz science and fulfill the lack of a high-power THz radiation source [1]. This kind of source injects a radio frequency (RF)-modulated electron beam whose bunch length is comparable to the radiation wavelength of an undulator. Electrons move with sinusoidal trajectory inside the undulator and emit radiation called coherent undulator radiation (CUR). The source is composed of few components and operates in a low-beam energy range. Consequently, it requires a low construction and operation cost, which is suitable for a small research laboratory. Currently, such sources have been developed in many laboratories, for example, FEL-CATs (0.4 to 0.7 THz) at ENEA Frascati [2] and the compact system at Peking University [3].

We are developing a new compact high-power THz radiation source at the Institute of Advanced Energy, Kyoto University [4]. In the first step of development, THz radiation is generated via CUR from a low-energy short bunch electron injected into a 10-period planar undulator. The expected CUR wavelength is in the range from 500 to 1700 μm (0.17 to 0.6 THz). A magnetic chicane bunch compressor is employed to compress the electron bunches to have the bunch length comparable to the radiation wavelength. The electron beam is generated from a 1.6-cell S-band BNL-type photocathode RF-gun [5] with a solenoid magnet for beam focusing and emittance compensation. This gun provides a maximum

beam energy of 4.6 MeV, and the beam properties were measured and reported in Reference [6]. A triplet quadrupole magnet upstream from the undulator is used for focusing the beam on the undulator. A rectangular bending magnet is used as the beam dump, placed at the end of the beamline. The total length of the system is 3.7 m and is located in the same accelerator hall as the Kyoto University Free-Electron Laser (KU-FEL) [7]. Both systems share an RF-power source and a picosecond UV laser system [8]. A schematic view of the system is shown in Figure 1.

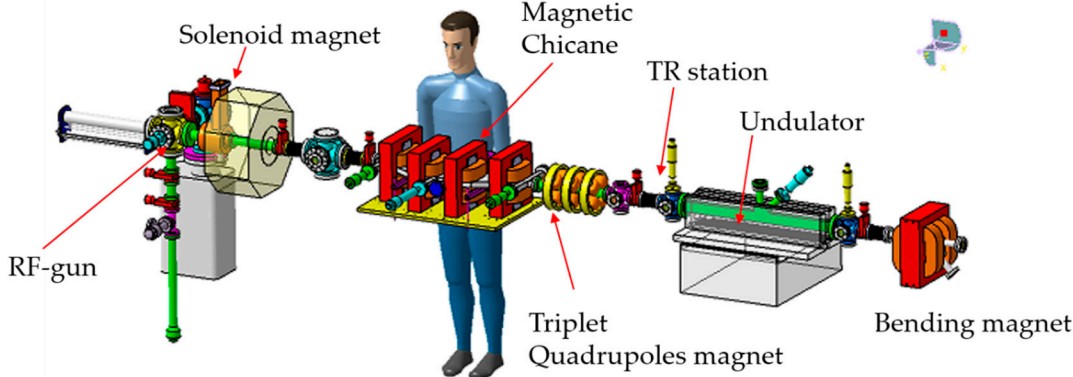

**Figure 1.** Schematic view of the completed beam line of the compact terahertz radiation source at Kyoto University.

The characteristics of the bunch compressor were studied numerically by using general particle tracer (GPT) code [9] and experimentally by observing the coherent transition radiation (CTR), which is generated by an electron bunch hitting a tilted 45-degree thin aluminum foil installed at the transition radiation (TR) station. The bunch length of the compressed bunch was estimated from the CTR spectra analyzed by an in-air Michelson interferometer. This method is called the frequency-resolved technique, was proposed by Barry et al. [10], and was first successfully applied by Lihn et al. [11]. The details of the bunch compressor, the measurement procedure, and the results are presented in this paper.

## 2. Magnetic Chicane Bunch Compressor

The magnetic chicane bunch compressor consists of four dipole bending magnets arranged in a longitudinal symmetric layout, as shown in Figure 2a. The magnets are H-type rectangular dipole electromagnets with a pole length of 65 mm, a pole width of 100 mm, and a pole gap of 30 mm. A $2 \times 5$ mm$^2$ rectangular cross-section air-cooled polyester enamel coated copper wire (PEW) with 190 turns is used as an electromagnetic coil for each pole. The maximum electrical current supplied to the coil is 12.1 A, which provides a maximum magnetic field of 193 mT. The distance between the center of each magnet in a longitudinal direction is 190 mm. The second and third magnets can be moved in a horizontal direction with a maximum offset from the center of the first magnet of 82 mm and a maximum beam deflection angle of 35 degrees, limited by the vacuum chamber. The chicane creates a dispersive beam: Inside that, a high-energy particle travels on a shorter path than a low-energy one does. This allows bunch compression if the electron bunch has a lower energy at the head and higher energy at the tail. The schematic of the chicane and the energy-time phase spaces of the maximum compression condition of the electron bunch at the entrance and exit of the chicane are shown in Figure 2b.

The minimum bunch length of the compressed beam at the chicane exit could be obtained by optimizing the correlation between the efficiency of the bunch compression and the energy chirp of the initial bunch. The efficiency of the chicane compression was determined by the first-order momentum compaction factor $R_{56}$, which is defined by $ds/d\delta$, where $ds$ is the total path length difference of electron trajectory and $d\delta$ is the relative energy spread. The energy chirp of the bunch depends on the energy modulation by the accelerating electric field $E(\phi)$ inside the RF gun and can be changed by adjusting

the timing of the laser pulse relative to the phase of $E(\phi)$. The relative phase between the zero crossing of $E(\phi)$ and the end of the laser pulse is defined as the laser injection phase $\phi_L$. A diagram of the definition of the laser injection phase and the energy-time phase space of the bunch at the RF gun exit are shown in Figure 3.

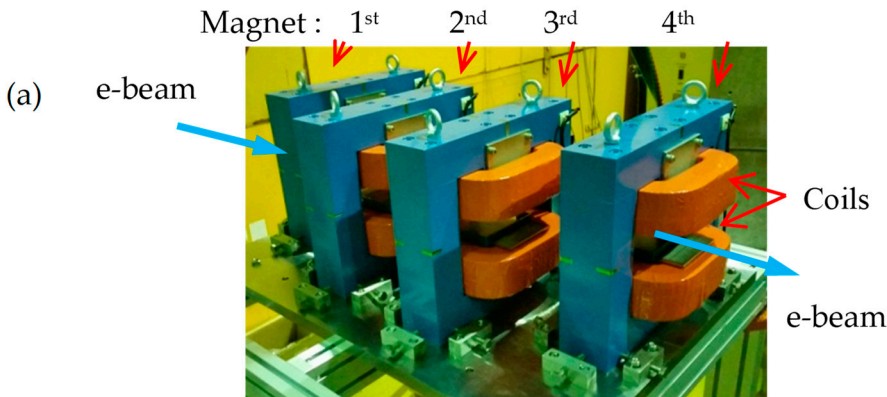

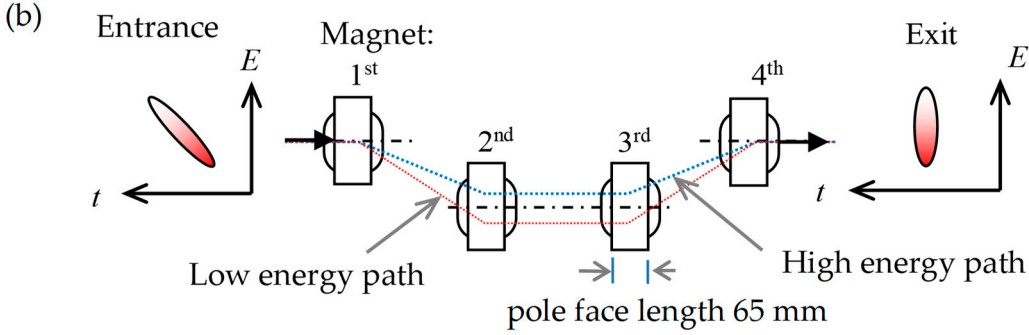

**Figure 2.** (**a**) Magnetic chicane bunch compressor (without vacuum chamber). (**b**) Schematic diagram of the top view of the chicane and the energy-time phase space of the electron bunch at the entrance and the exit of the chicane at the maximum bunch compression condition. The head of the bunch is represented by the darker shade.

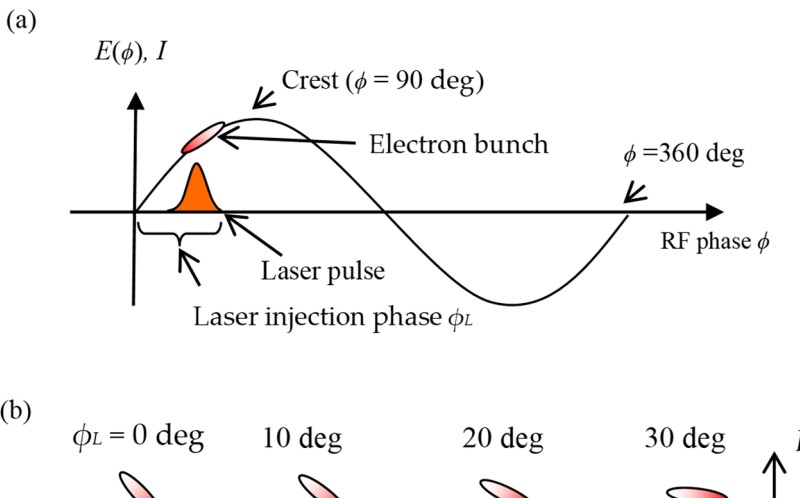

**Figure 3.** (**a**) Definition of the laser injection phase $\phi_L$. (**b**) Energy-time phase space at the RF-gun exit of an electron bunch for different laser injection phases. The head of the bunch is represented by the darker shade.

The $R_{56}$ of the magnetic chicane was calculated at the nominal beam energies of 4.6 MeV by GPT using the built-in magnet element, including the fringe fields. The calculated $R_{56}$ and the measured magnetic field at the center of the pole gap of the bending magnet as a function of the chicane excited current is shown in Figure 4.

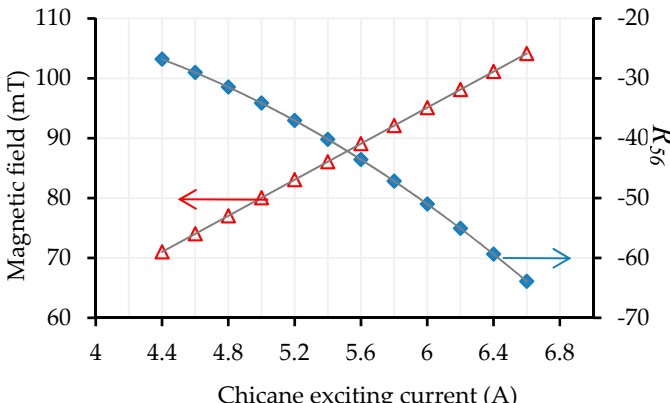

**Figure 4.** Measured magnetic field and the $R_{56}$ calculated by a general particle tracer (GPT) as a function of chicane excited current.

## 3. Coherent Transition Radiation

Transition radiation occurs when electrons cross a boundary of materials with different dielectric constants. Electrons emit radiations in a hollow cone shape in both forward and backward directions. For a long bunch length electron beam, each electron emits radiation randomly, resulting in incoherent TR whose intensity is proportional to the number of electrons. In the case of a short bunch, when a bunch length is comparable to the radiation wavelength, the superposition of TR becomes a CTR whose intensity is much higher than the incoherent radiation, and the intensity is proportional to the number of electrons squared. The total CTR intensity ($I_{CTR}$), which is emitted by a bunch of $N_e$ electrons [12], can be calculated by

$$I_{CTR}(\omega) = N_e^2 \, I_{TR} \, |h(\omega)|^2, \tag{1}$$

where $I_{TR}$ is the transition radiation generated by a single electron, and $h(\omega)$ is the Fourier transform of the electron distribution function or the form factor. Here, $h(\omega)$ is defined by

$$h(\omega) = |\int e^{ikz} h(z) \, dz|^2, \tag{2}$$

where $k$ is the wave number of the radiation, $k = \omega/c = 2\pi/\lambda$, $h(z)$ is the normalized distribution function of electrons in the longitudinal direction, and $z$ is the longitudinal position of electrons. In this study, we used a Michelson interferometer to measure the autocorrelation intensity. The CTR entering the interferometer is split into two beams by a beam splitter. One beam is reflected off a fixed mirror, while another one is reflected back by a movable mirror that creates a path length difference. After that, both beams are combined and generate the autocorrelation radiation. The intensity of the radiation dependent on the path difference or the interferogram $I(\delta)$ can be represented by [12]

$$I(\delta) \propto 2 \, \text{Re} \int |RT|^2 \, E(t) \, E^*(t+\delta/c) \, dt, \tag{3}$$

where $R$ and $T$ are the reflection and transmission coefficients of the beam splitter, $E(t)$ and $E^*(t+\delta/c)$ are the time-varying electric fields reflected from the fixed and movable mirrors, $t$ is the time, $\delta$ is the

path difference, and $c$ is the speed of light. The Fourier transform (FT) of the interferogram gives the power spectrum $I(\omega)$ of the radiation

$$I(\omega) = \text{FT} \{I(\delta)\} \propto \left| R(\omega) \, T(\omega) \, \widetilde{E}(\omega) \right|^2 \propto \left| h(\omega) \right|^2, \tag{4}$$

where $R(\omega)$, $T(\omega)$, $\widetilde{E}(\omega)$ are the reflection coefficient, the transmission coefficient, and the electric field in the frequency domain. In an actual measurement, the power spectrum is affected not only by the efficiency of a beam splitter but also by the frequency dependence of optical components such as lenses, mirrors, and the medium. The diffraction also distorts the power spectrum, which is included in $\widetilde{E}(\omega)$. These factors have to be carefully considered in an analysis of the measured CTR spectrum.

## 4. Measurement Set-Up

The beamline for the CTR experiment is shown in Figure 5. The centers of the magnetic chicane and the CTR station were located at a distance of 1170 mm and 2130 mm from the exit of the RF-gun, respectively. A copper photocathode was illuminated by picosecond UV laser pulses with a wavelength of 266 nm. The laser had a repetition rate of 89.25 MHz, and a pulse duration at the full width at half maximum (FWHM) of around 6 ps [13]. The CTR radiator used an 11-μm thick aluminum foil arranged at 45 degrees relative to the beam direction, and a fluorescence screen was attached behind the aluminum foil to observe the beam position. Actually, the fluorescence screen also generated backward CTR, but it was blocked by the aluminum foil. The CTR was extracted out of a vacuum through a 4-mm-thick Z-cut natural crystal quartz window (Fuji Ideck, TF70, Ibaraki, Japan) with an aperture diameter of 33 mm.

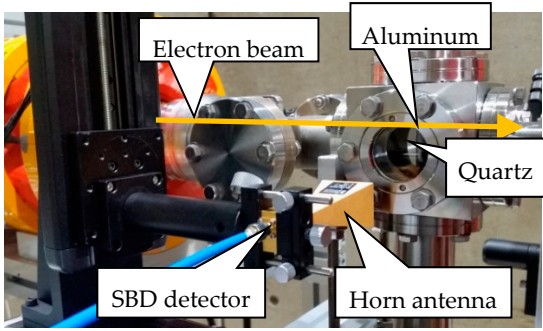

**Figure 5.** Setup of the CTR spatial distribution measurement.

Two small-size uncooled detectors were used for measuring the CTR intensity. First, a Schottky-barrier zero bias diode (SBD) detector (Millitech, DET-05-RPFW0, Northampton, MA, U.S.A.) operated in the G-band frequency range (140 to 220 GHz) and had a sensitivity of 250 mV/mW [14]. The SBD detector could pick the signal in one direction of polarization at a time because it was installed in a rectangular waveguide. Second, a pyroelectric detector (PHLUXi, PYD-1-018, Miyagi, Japan) equipped with a built-in lens and a visible light filter operated in a broad frequency range [15]. Unfortunately, the calibration of these detectors in the THz radiation region has not been performed yet. The signals from both CTR detectors were recorded by a 1-GHz bandwidth oscilloscope (Tektronix, DPO4104, Beaverton, OR, U.S.A.). A 10-cm cubic carbon placed in air at the end of the beamline was employed as a Faraday cup for the bunch charge measurement. The electron beam passed through a 0.2-mm-thick copper window, and the signal from the Faraday cup was read by a 200-MHz bandwidth oscilloscope (Iwatsu, DS-5624A, Tokyo, Japan). The input impedances of both scopes were set at 1 MΩ. The magnet power supplies, the actuators, the translational stages, the cameras, and the waveform measurement instrument were controlled under a LabVIEW environment. The parameters for the experiment are listed in Table 1.

**Table 1.** System and beam parameters for the coherent transition radiation (CTR) experiment. FWHM: Full width at half maximum.

| Parameters | Value |
|---|---|
| Beam energy | 4.6 MeV |
| RF power for the RF-gun | 9 MW |
| RF frequency/RF pulse duration Number of RF micropulses per macropulse | 2856 MHz/2 μs 5700 |
| 1st-order momentum compaction factor ($R_{56}$) | min−68.7 mm |
| Solenoid magnetic field | 194 mT |
| Laser repetition rate of electron micropulse | 89.25 MHz |
| Laser pulse duration (FWHM) | ~6 ps |
| Laser temporal/spatial distribution | Gaussian/Gaussian |
| Laser size at cathode (horizontal/vertical) | 0.4/0.5 mm (rms) |
| Number of laser pulses per macropulse | 1 to 4 pulses |
| Laser pulse energy per macropulse | 133 to 300 μJ |

### 4.1. Spatial Distribution of the CTR

The intensity mapping of the spatial distribution of the CTR was performed by scanning the SBD diode detector across the axes. A 25-dB gain pyramidal horn antenna (Ducommun, ARH-1025-02, Santa Ana, CA, U.S.A.) was attached to the detector to obtain higher detected CTR signals. The detector was mounted on two translational stages (Sigmakoki, SGSP20-85, Tokyo, Japan) and was scanned at 2-mm intervals in the horizontal and vertical directions. The distance from the CTR radiation point to the SBD detector was approximately 135 mm, and the set-up is shown in Figure 5.

### 4.2. CTR Intensity

The total intensity of the radiation was measured by using three gold-coated mirrors, which were two 2-inch off-axis parabolic mirrors (Thorlabs, MPD249-M-01, $f$ = 106.5 mm, Newton, NJ, U.S.A.), and a flat mirror (Thorlabs, PF20-03-M0, Newton, NJ, U.S.A.) to transport and focus the CTR onto the detector. The focal point and the detector center were aligned by scanning the detector in two dimensions in the normal plane. The readout signals were averaged over 10 measurements. The set-up of the CTR intensity measurement is shown in Figure 6.

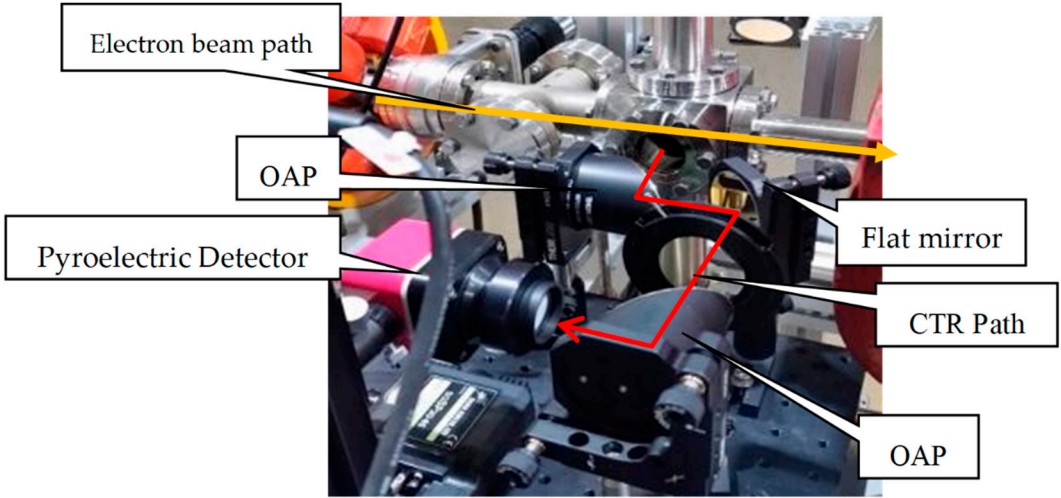

**Figure 6.** Setup of the CTR intensity measurement. OAP indicates off-axis parabolic mirrors.

### 4.3. CTR Spectrum

The CTR spectra were analyzed by an in-air Michelson interferometer using a pyroelectric detector for measuring the autocorrelation intensities. The beam splitter was a 2-inch in diameter sapphire substrate with a thickness of 100 μm. The theoretical calculation of the magnitude of transmittance

$T(\omega)^2$, the reflectance $R(\omega)^2$, and the efficiency $|T(\omega)R(\omega)|^2$ of the sapphire beam splitter as a function of frequency [16] are shown in Figure 7. The fixed and movable mirrors were 2-inch gold-coated flat mirrors (Thorlabs, PF20-03-M01, Newton, NJ, U.S.A.). The movable mirror was mounted on the translational stage (Sigmakoki, OSMS20-85, Tokyo, Japan) and moved at a 0.1-mm interval equivalent to an optical path difference of 0.2 mm and a sampling frequency of 1.5 THz. The total moving length of the mirror was 7 mm, corresponding to a total path difference of 14 mm and a time window of 46.7 ps. Therefore, the frequency resolution, which was the inverse of the time window, was 21.4 GHz. The lead blocks were placed beside the beamline to reduce the background X-ray events. The set-up of the Michelson interferometer is shown in Figure 8.

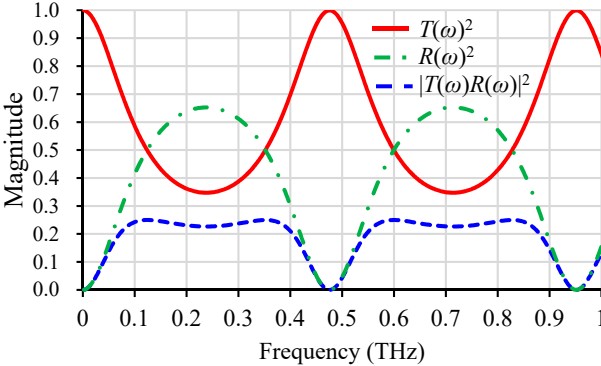

**Figure 7.** Theoretical calculations of the magnitude of Transmittance $T(\omega)^2$, reflectance $R(\omega)^2$, and efficiency $|T(\omega)R(\omega)|^2$ of the sapphire beam splitter as a function of frequency.

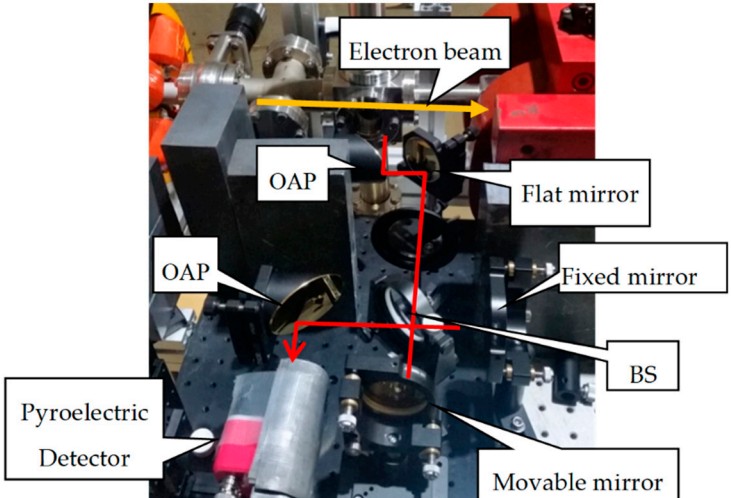

**Figure 8.** Set-up of the Michelson interferometer. The red lines represent the CTR path, OAP is the off-axis parabolic mirror, and BS is the beam splitter.

## 5. Results and Analysis

### 5.1. Spatial Distributions of the CTR

Figure 9 shows the intensity mapping of the CTR in vertical and horizontal polarizations. The measurement was performed at a laser pulse energy of 133 μJ and 4 pulses per macropulse. The results clearly show the lobe structure characteristic of CTR, and the distance between peaks was about 30 mm, which was equal to the open angle of 6.3 degrees, which corresponded to the inverse of the Lorentz factor, $1/\gamma$, at 5.7 degrees.

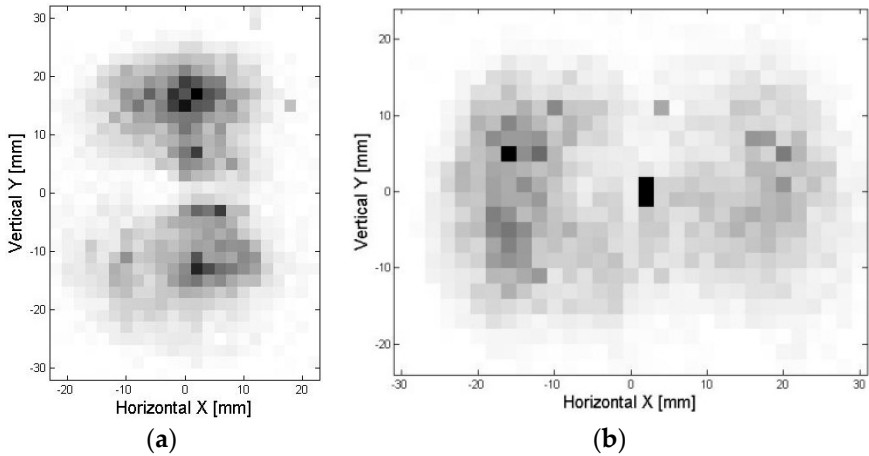

(**a**)                                    (**b**)

**Figure 9.** Intensity mapping of the CTR in (**a**) vertical and (**b**) horizontal polarizations. Note that the dark pixels in the horizontal polarization occurred in the measurement during RF breakdown.

*5.2. Bunch Compression Characteristics*

In this section, we investigate the correlation between the energy-time phase space of the initial bunch and the $R_{56}$ of the bunch compressor by varying the laser injection phase at a given $R_{56}$ value. The laser injection phase was varied from 0 to 110 degrees while the chicane was turned off and on, which provided $R_{56}$ values of 0 and $-26.8$ mm. Note that this $R_{56}$ value was not the optimal value for the maximum bunch compression condition. The optimal one was investigated and is explained in Section 5.3. The number of the laser pulses and the laser energy per macropulse were 4 pulses and 300 µJ, respectively. The measurement results of the bunch charge and the CTR intensity as a function of the laser injection phase are shown in Figures 10 and 11, respectively.

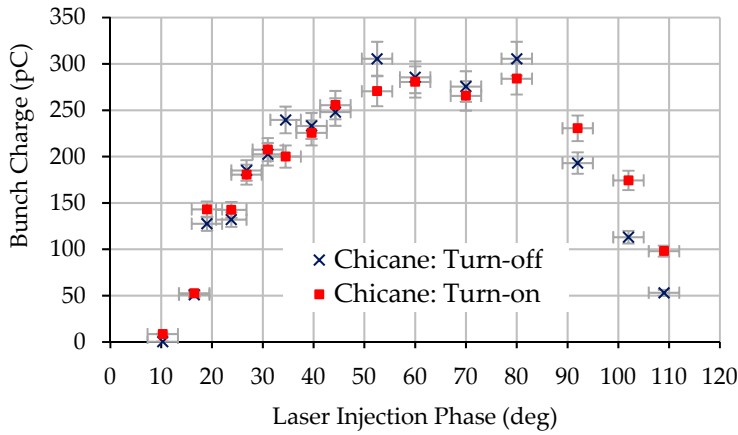

**Figure 10.** Bunch charges as a function of the laser injection phases at an $R_{56}$ of $-26.8$ mm.

The results showed that the bunch charge gradually increased from zero at the laser injection phase of zero degrees until a peak charge of 280 pC at a phase of around 50 to 75 degrees, and decreased at the higher phases. This was approximately the same as shown in Figure 10, meaning that no electrons were lost in the bunch compression section. Figure 11 shows that the CTR intensity was very sensitive to the laser injection phases, especially when the chicane was turned on. The chicane successfully compressed the bunch at a phase less than 45 degrees, where the CTR intensities were enhanced. The peak CTR intensity occurred at the laser injection phase of around 20 to 30 degrees. In the case of the chicane being turned off, this means that the bunch length of this phase range was shorter than the other phase. When the chicane was turned on, the energy chirp of the bunch of this phase range was suitable for bunch compression. The peak CTR intensity could be enhanced up to a

factor of 2.6 compared to the turned-off chicane. At a laser injection phase of over 45 degrees, the bunch head had higher energy than the bunch tail due to the acceleration behind the RF-crest. Such a bunch expanded longitudinally when it passed through the chicane, resulting in the lower CTR intensity.

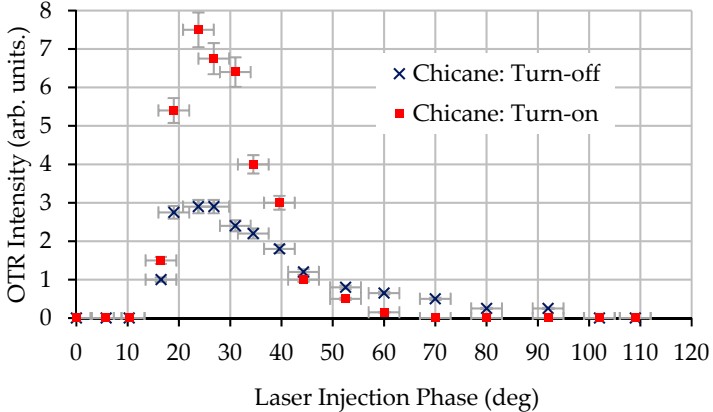

**Figure 11.** The measured CTR intensity dependence on the laser injection phases with the chicane turned off and on ($R_{56}$ = 0, −26.8 mm).

### 5.3. Maximum Bunch Compression Condition

The maximum bunch compression condition was investigated by the power spectrum of CTR calculated by fast Fourier transform of an interferogram, as described in Section 3. The interferograms were measured by Michelson interferometer, and the measurements were performed by varying the $R_{56}$ from −31.5 to −68.7 mm, while the laser injection phase was fixed at 30 degrees. The laser energy was 300 µJ with 4 pulses per macropulse, and the solenoid magnetic field was fixed at 194 mT. Only the CTR signals of the compressed bunch were measured, because their signals were strong enough to be detected. The measured interferograms are shown in Figure 12. Note that the interferograms in Figure 12 were shifted vertically for ease of seeing the lines.

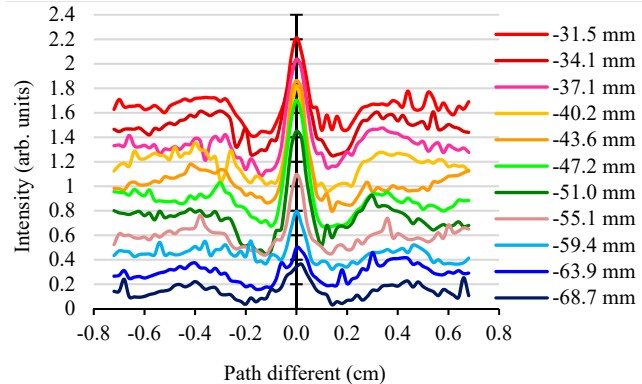

**Figure 12.** Measured interferograms for different $R_{56}$ values at solenoid magnetic fields of 194 mT and a laser injection phase of 30 degrees.

The fast Fourier transform corresponding to the measured interferograms or the CTR spectra are shown in Figure 13. Most of the CTR spectrum contained frequency components lower than 0.25 THz, and the peak of the spectrum was around 0.08 THz. However, these spectra included frequency dependency of all optical components of the measurement system and media, for example suppression in the low-frequency region due to a drop in beam splitter efficiency (see Figure 7). The diffraction limited spectral transmission due to limited aperture and absorption by the water vapor in air at a frequency of 0.56 THz [17]. The maximum compression condition seemed to occur at an $R_{56}$ of −43.6 mm, because its frequency-integrated power spectrum was highest.

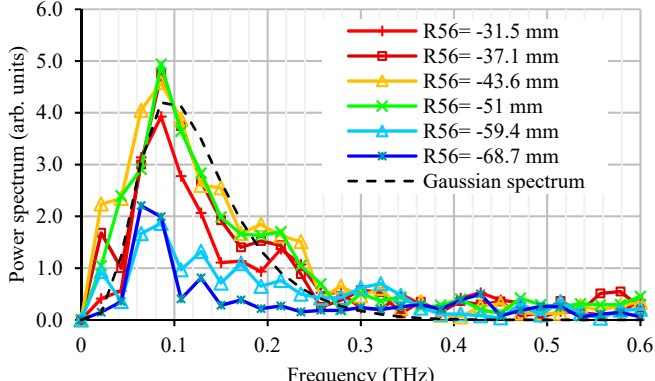

**Figure 13.** Power spectra of the CTR with different $R_{56}$ values calculated from the interferograms in Figure 12. They were compared to the spectrum of a Gaussian distribution bunch with an rms bunch length of 1 ps without any bunch compression (dashed line).

### 5.4. Bunch Length Estimation

In this study, the rms bunch length was roughly estimated by fitting the measured spectrum to the spectrum of a known particle distribution. We assumed that the longitudinal bunch distribution was a Gaussian distribution whose power spectrum was $Ce^{-\omega^2\sigma^2}$, where $C$ is a constant, $\omega$ is the angular frequency, and $\sigma$ is the rms bunch length. The power spectrum of the Gaussian distribution bunch contained continuous frequency components with the maximum intensity of $C$ at zero frequency and smaller components toward higher frequencies. The suppression of the power spectrum intensity at a low frequency was considered only by the spectral efficiency drop of the sapphire beam splitter. The fitting of the measured CTR spectra was performed by the least squares method at a frequency range higher than 0.08 THz, where the spectra had less distortion from the drop in the beam splitter efficiency. The example of the fitting CTR spectrum is shown in Figure 14. The rms bunch length of the fitted CTR spectra as a function of $R_{56}$ is shown in Figure 15. The minimum bunch length was less than 1 ps, with $R_{56}$ around $-45$ mm. Note that the error in the rms bunch length was due to the curve fitting processes.

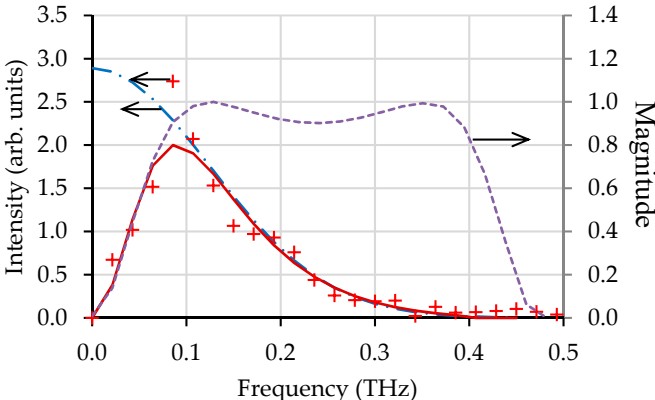

**Figure 14.** Measured CTR spectrum (cross marks), the fitting Gaussian spectrum (dot-dashed line), the normalized beam splitter efficiency (dashed line), and the fitting CTR spectrum of the Gaussian distribution bunch (solid line).

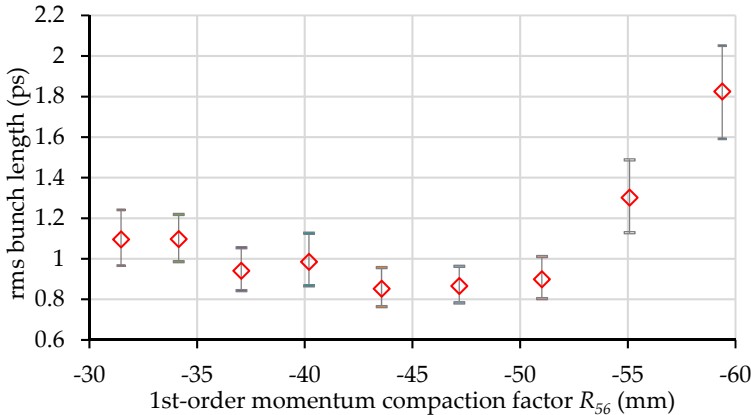

**Figure 15.** Estimated rms bunch length by fitting the measured CTR spectrum to the Gaussian power spectrum.

## 6. Conclusions

The magnetic chicane bunch compressor for a compact THz radiation source was designed, installed, and commissioned. The characteristics of the chicane and the compressed electron beam properties were investigated using the coherent transition radiation technique. Even with a low beam energy of 4.6 MeV, the chicane was able to compress the electron beam at a laser injection phase less than 45 degrees. CTR intensity enhancement by the bunch compressor was at a maximum at a laser injection phase of around 24 degrees, and the enhancement factor was up to 2.6. The CTR spectra were measured by a Michelson interferometer and had a frequency lower than 0.25 THz, with a peak at 0.08 THz and a smaller peak at a frequency of 0.2 THz. The optimum compression of the present beam condition was obtained at a first-order momentum compaction factor of around $-45$ mm, corresponding to the excited current of the chicane of 5.7 A. This condition gave the estimated rms bunch length by fitting with the power spectrum of the Gaussian distribution bunch less than 1 ps. However, the measured CTR spectra showed that the bunch distribution was not perfectly a Gaussian distribution and contained a complex structure inside. Further investigation of the bunch profile reconstruction would bring more information on the electron bunch distribution. Finally, due to the magnetic chicane, the electron bunch was successfully compressed. We expect that the THz radiation power generated from the undulator was in the order of megawatts.

**Author Contributions:** Methodology, H.Z.; writing—original draft, S.S.; writing—review and editing, H.Z., T.K., and H.O.

**Funding:** This work was supported by MEXT/JSPS KAKENHI Grant Number 26706026.

**Conflicts of Interest:** The authors declare no conflicts of interest.

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
