# Peer review of "Investigation of Bunch Compressor and Compressed Electron Beam Characteristics by Coherent Transition Radiation"

_2571-712X, doi:10.3390/particles2010003_

Round 1

Reviewer 1 Report

Fig. 7: please describe in the context how you derived/obtained the curves for transmittance and reflectance.

Fig. 11: please explain in the context why there also exists a peak of the CTR intensity around 25 degree when the chicane was turned off.

Fig. 13: the difference between the high-frequency components of R56=-43.6mm and -51mm seems insignificant. It is a little bit weak to state that "its CTR spectrum contains high frequency components more than the others" in Line 294-295. Can use, e.g., frequency-integrated power instead. 

Line 309, Fig. 15, et al.: In your case you should have nonlinear compression, and the beam profile should be rather different compared to Gaussian. Therefore the pulse duration derived herein might be rather rough. It would be better if you can analyze the error based on the comparison between Gaussian function and beam profile from, e.g., particle tracking. 

Author Response

Dear Reviewer 1

Thank you for comments on my manuscript.

I have response your comments point by point.

Thank you

Best regards

Sikharin

Reviewer 2 Report

Fig. 4: Red and blue lines need to be labeled.  It is not clear what the arrows are for or why there are trend line equations.  The arrows should be removed and the authors should assess whether the trend line equations are necessary.

236 lope -> lobe

255-256 "had almost no different that mean the electrons did not lose in the bunch compression section." -> "was approximately the same as shown in Fig. 10, meaning that no electrons were lost in the bunch compression section"

293-295: "The maximum compression condition occurred at the R56 of -43.6 mm because its CTR spectrum contains high frequency 295 components more than the others."
I assume that this conclusion is based on the local maximum in the THz signal near 0.2 THz.  If so state this.  Otherwise, clarify how this was determined.

Finally, I think it would benefit the paper if some additional conclusions were clearly stated.  To what extent does the THz production match expectations and/or requirements?  Does the measured bunch length of Fig. 15 length agree with theory?  If not, have the authors identified potential causes and ways to improve their setup?

Author Response

Dear Reviewer 2

Thank you for your valuable comments on my manuscript.

I response your comments point by point.

Thank you

Best regards

Sikharin

Reviewer 3 Report

I think it's an interseting paper, but I propose a revision. The basis are explained.

E.g. Errorbars are missing in all measurements! 

scientific language sould be improved / slang should be avoided, most figures subtitles have to be improved.

You show the charge distribution along the phase but not the momentum. Did you implemented the momentum change with the phase?

The setups are partially described in very detail but I don't understand the important things, e.g. how one switch changes betweeen the setup decribed in 4.1., 4.2. and 4.3.  I guess, they are mounted one after each other?

Author Response

Dear Reviewer 3

Thank you for comments on my manuscript.

I have response your comments point by point.

Thank you

Best regards

Reviewer 4 Report

Some comments and recommendations for the manuscript particles-390266

22 Nov. 2018

This article reports on measurement results and optimizations of coherent THz radiation (CTR) generation with a compact linear accelerator (linac)-based radiation source at the Kyoto University. The results are based on the detailed measurements depending on the laser injection phase and the 1st-order momentum compaction factor of the bunch compressor chicane. 

The compact linac comprises a photocathode RF gun, a bunch compressor chicane and an undulator. The electron beam energy is 4.6 MeV.  The typical RF power supplied for the RF gun is 9 MW. The CTR intensity was measured by a pyroelectric detector. The CTR spatial distribution was measured by a Schottky-barrier zero bias diode with a horn antenna. The bunch length was measured by a Michelson interferometer with pyroelectric detector. The CTR intensity was optimized and maximized as functions of the laser injection phase and the momentum compaction factor. The optimized bunch length was obtained to be 0.85 ps at the momentum compaction factor of -43.6 mm and the laser injection phase of 24 degrees, where the beam charge was ~150 pC with the chicane turn-on. 

I believe that this article may give an important step to realize a compact THz-radiation source towards high power and tunable radiation sources. Thus, this article is well worth publishing without any essential corrections in the contents. It seems that the article can be easily understood due to plain English.

However, there are some insufficient and unclear descriptions along with many minor mistakes and desirable revisions, while the obtained results are very interesting. Some minor amendments are also suggested.  Thus, as an interested reader of this article, I will give the following minor amendments along with a few comments.

Comments and suggested points

(1)  In the lines 172-173 of section 4, you mention that as a CTR radiator a thin aluminum foil is used and a fluorescence screen is attached to the back face of it. In such a situation, CTR signals are generated necessarilyfrom both the surfaces of the aluminum foil and also the fluorescence screen. How did you suppress transition radiation from the fluorescence screen? It should be described and clarified.

(2)  In the lines 181-183 of section 4, you mention that the calibration of the detector in the THz radiation region has not be performed yet. Does this sentence mean that the calibrations for both a pyroelectric and SBD detector or for only one detector? In a case of only one detector calibration, which detector was calibrated? It should be clarified. 

(3)  In the lines 182-183 of section 4, you mention that the signals from the CTR detectors were recorded by a 1 GHz bandwidth oscilloscope (…). Does this sentence mean that the signals from both pyroelectric and SBD detectors were recorded or only one detector signal was recorded? It should be clarified. 

(4)  In Fig. 10, there is a clear dip structure around 70 degrees in the bunch charge distribution as a function of the laser injection phase. Was such a dip structure caused by the measurement issue or beam dynamics? It is better to describe the reason briefly.  I think that it seems that the dip structure was caused by the measurement issue.

(5)  In Introduction, you mention that “they have the potential to produce a high power and tunable radiation”. However, the authors do not describe the power level of this accelerator-based THz radiation source anywhere, because the power calibration has not been finished yet. However, it is better to describe the power level at least in Conclusion. I think that it may be enough to describe the expected power value, if you do not have any precise calibration results.

Minor revisions required 

·In the line 18 of Abstract, it is better to use the plural form in unit for … 45 degrees.

·In the line 19 of Abstract, it is better to use the plural form in unit for … 24 degrees.

·In the line 69 of section 2, it seems that there is an extra space between the words “current” and “supplied” in “.. electrical current supplied to…”, and it is better to delete it.

·In the figure caption of Fig. 2, it is better to add the axis legends in the longitudinal phase space drawings, that is, “t” for the horizontal axis and “E” for the vertical axis.

·In Fig. 2, it can be seen the length of 65 mm and the arrows which indicate the length section. However, it is not clear what the length section means. It maybe shows the length including the pole and coil lengths. It should be clarified.

·In Fig. 2, the longitudinal phase space drawings are used with a red color gradation, and however, it cannot be found any explanations about it. However, in the figure caption of Fig. 3, we can find some explanation on it, that is, “The head of the bunch represents….”. If this sentence can be applied to Fig. 2, it is better to add this sentence again in the figure caption.

·In Fig. 3 (b), an undefined axis legend is used in the longitudinal phase space drawings, that is, “gamma” for the vertical axis. It is better to use “E” instead of “gamma” because the gamma is not defined anywhere.

·In Fig. 4, we cannot entirely see the fitting curves because of the faint colors. If you want to emphasize the results of the fitting curves, it should be emphasized by using some clear solid lines. And we cannot also see any explanations on the results of the fitting function formula, which are shown in the figure. It should be clearly explained in the figure caption.

·In the lines 145 and 146 of section 3, it is better to revise the sentence such as “One beam is reflected by…. While another one is reflected back by….”.

·In eqs. (4) and (5) of section 3, we can see the letter, which seems to indicate “alpha” instead of “proportional”. 

·In the line 154 of section 3, it is better to revise such as “The Fourier transformation (FT) of the ……” because in eq. (5) we cannot see any definitions on FT.

·In the lines 160-161 of section 3, it is better to revise such as “…, the power spectrum is affected not only by the efficiency… but also by the frequency dependence of….”.

·In the line 169 of section 4, it is better to revise such as “by the picosecond UV laser pulses…..”.

·In the line 182 of section 4, it is better to revise such as “… has not been performed yet”.

·In the title of subsection 4.1, it is better to revise such as “Spatial Distribution of the CTR”.

·In the line 205 of section 4, it is better to revise such as “… (Thorlabs, ….), and ….”.

·In Fig. 6 of section 4, in the figure, it is not clear “Electron beam” and “OAP”. Here, OAP is not defined anywhere. It should be defined in the figure caption. For example, OAPs indicate the off-axis parabolic mirrors.

·In the figure caption of Fig. 6, it is better to delete the parenthesis in “(The red line ….)”. 

·In the line 217 of section 4, we can find a mistype, “the efficiency”.

·In Fig. 7 of section 4, does the results show experimental one or catalog-specified one or calculated results? It should be clarified.

·In the title of subsection 5.1, it is better to revise such as “Spatial Distributions of the CTR”.

·In the line 237 of section 5, it is better to revise such as “the open angle of 6.3 degrees”. 

·In the line 238 of section 5, it is better to revise such as “…., which corresponds to the inverse of the Lorentz factor, …., at 5.7 degrees”.

·In the figure caption of Fig. 9 of section 5, it is better to revise “…that the dark pixels in the ……. occurred in the measurement during….”.

·In the line 250 of section 5, unclear section number is found, that is, “the subsection D”. Is it right that this is subsection 5.3? 

·In the line 251 of section 5, it is better to revise such as “The number of …… were 4 pulses and 300 microJ, respectively.”. The order is wrong in the sentence.

·In the line 252 of section 5, it is better to revise such as “… are shown in Figs. 10 and 11, respectively.”.

·In the line 254 of section 5, it is better to revise such as “until the peak charge of…”.

·In the line 258 of section 5, it is better to revise such as “…less than 45 degrees…”.

·In the lines 255-256 of section 5, it is better to revise such as “There had no difference on the bunch charge with the turn-off and -on of the chicane, that means the electrons did not lose in the bunch compression section”.

·In the lines 261-262 of section 5, it is better to revise such as “Such a bunch is expanded longitudinally when it passes through….”.

·In the figure caption of Fig. 10, a period is required at the end of the sentence. 

·In the figure caption of Fig. 11, it is better to revise such as “… the laser injection phases with the chicane turn-off and -on (R56….).”. 

·In the line 271 of subsection 5.3, the section number II is wrong. The right is “section 2”. Please check the section number.

·In the line 273 of subsection 5.3, it is better to use “… at 30 degrees”.

·In the line 276 of subsection 5.3, there is a mistype, that is, “Fig. 12”.

·In the figure caption of Fig. 12, it is better to use “… of 30 degrees”.

·In the figure caption of Fig. 13, it is better to revise “Power spectra of the CTR…”.

·In the figure caption of Fig. 13, it is better to revise such as “Power spectra of the CTR…. Fig. 12. They are compared with the spectrum of Gaussian….without any bunch compression (dashed line).”. In this sentence, I give you a case without any bunch compression. However, it is not clear that the bunch compression was applied (or not) for the gaussian bunch. It should be clarified.

·In the line 292 of subsection 5.3, it is better to insert one space, that is, “Fig. 7”.

·In the line 300 of subsection 5.4, it is better to remove an extra period, that is, “Ce^(-w^2s^2)”.

·In the line 307 of subsection 5.4, it is better to insert one space between the sentences, that is, “… in Fig. 14. The rms….”.

·In the figure caption of Fig. 14, it is better to insert a word such as “…. and the fitting CTR spectrum of the Gaussian distribution bunch (solid line).”.

·In Fig. 14, it is better to insert the axis legend and also unit in the righthand side. It should be clarified. If you insert the axis legend, it should be clarified the correspondence of the curves to the vertical axes.

·In the line 328 of section 6, it is better to revise such as “24 degrees”. 

·In the line 331 of section 6, it is better to revise such as “the 1st-order momentum compaction factor of …”. 

·In references (the line 343), it is better to add a period at the end of line of 2.

·In references (the line 360), it is better to add a period at the end of line of 11.

·In references (the line 365), it is better to add a period at the end of line of 14.

·In references (the line 366), it is better to add a period at the end of line of 15.

·In references (the line 367), there are some mistypes, that is, “Measurement” and “the transmission”.

Author Response

Dear Reviewer 4

Thank you for giving your valuable time for checking and comment my manuscript.

I have responded to your comments point by point and send it to you.

Thank you

Best regards

Sikharin

Round 2

Reviewer 3 Report

Even the statistical errors have not been measured. The bunch length measurement is meaningless without errorbars! Actually I would guess the systematic errors are much higher than the statistic errors. The statistic error you still can determine! So give an estimate of the errorbars that one can judge the measurement! E.g. Fig. 15. you need to judge the errors from the transfer function. Otherwise, it’s not clear if the difference between the rms bunch length for different R56 is smaller than the measurement error! Afterwards a fit is possible to determine the minimum optimum R56 (-45deg +/- 5deg ???). I guess you have a GPT simulation of the bunch length as a function of the R56 value?  If yes, add it. You wrote in Fig. 10: “The drop of the bunch charge at the laser injection 263 phase of 70 degrees seems to be caused by the fluctuation in the measurement.” Such an explanation should not be necessary, therefore errorbars are required!

Author Response

Dear Reviewer 3 

Thank you for your comments.

I response your comments point by point.

Thank you 

Best regards 

Sikharin
